# Effect of *Tecoma stans* (L.) Juss. ex Kunth in a Murine Model of Metabolic Syndrome

**DOI:** 10.3390/plants11141794

**Published:** 2022-07-07

**Authors:** Dulce Lourdes Morales-Ferra, Miguel Ángel Zavala-Sánchez, Enrique Jiménez-Ferrer, Manasés González-Cortazar, Alejandro Zamilpa

**Affiliations:** 1Doctorado en Ciencias Biológicas y de la Salud, División de Ciencias Biológicas y de la Salud, Universidad Autónoma Metropolitana (UAM), México City 04960, Mexico; lou_ferra@hotmail.com; 2Centro de Investigación Biomédica del Sur, Instituto Mexicano del Seguro Social (IMSS), Argentina 1, Centro, Xochitepec CP 62790, Mexico; enriqueferrer_mx@yahoo.com (E.J.-F.); gmanases@hotmail.com (M.G.-C.); 3Departamento de Sistemas Biológicos, División de Ciencias Biológicas y de la Salud, Universidad Autónoma Metropolitana (UAM), México City 04960, Mexico

**Keywords:** metabolic syndrome, hypercaloric diet, *Tecoma stans*, luteolin, apigenin, chrysoeriol

## Abstract

Metabolic syndrome is a constellation of abnormalities related to insulin resistance with an unfortunately high prevalence worldwide. *Tecoma stans* (L.) Juss. Ex Kunth. is a well-known medicinal plant that has been studied in several biological models related to diabetes mellitus. The aim of this study was to evaluate the effects of *T. stans* on a hypercaloric diet-induced metabolic syndrome model. An organic fraction obtained using liquid–liquid separation from the hydroalcoholic extract of *T. stans* and four subfractions of this organic fraction were administered for ten weeks to C57BL6J male mice previously fed with a hypercaloric diet. The hypercaloric diet caused changes in glucose levels (from 65.3 to 221.5 mg/dL), body weight (31.3 to 42.2 g), triglycerides (91.4 to 177.7 mg/dL), systolic (89.9 to 110.3 mmHg) and diastolic (61.6 to 73.7 mg/dL) blood pressure, and insulin resistance (4.47 to 5.16). Treatment with *T. stans* resulted in improvements in triglycerides (83.4–125.0 mg/dL), systolic blood pressure (75.1–91.8 mmHg), and insulin resistance (4.72–4.93). However, the organic fraction and hydroalcoholic extract produced a better response in diastolic blood pressure (52.8–56.4 mmHg). Luteolin, apigenin, and chrysoeriol were the major constituents in the most active subfractions. Treatment with *T. stans*, particularly a luteolin-rich organic fraction, achieved an improvement in metabolic syndrome alterations.

## 1. Introduction

Metabolic syndrome (MS) is strongly related to biochemical, physiological, and anthropometric abnormalities produced by insulin resistance (IR) with high prevalence around the world [1]. MS is characterized by well-established traits: IR, obesity, pro-inflammatory state, atherogenic dyslipidaemia, arterial hypertension, and prothrombotic state [2], and increases the risk of developing diabetes mellitus (DM), cardiovascular disease, and cancer [3,4]. 

The most widely accepted hypothetical mechanism of the underlying pathophysiology of MS is IR with fatty acid hyperflux. Other possible mechanisms include low-grade chronic inflammation and oxidative stress [5]. Chronic exposure to supraphysiological glucose levels, such as those commonly found in MS, can also lead to the damage of insulin-sensitive tissues and pancreatic β cells [6], a process called glucotoxicity. 

Current therapeutic options for MS are limited to individual treatments for hypertension, hyperglycaemia, hypertriglyceridaemia, and IR [5]. Considering that increased triglyceride (TG) synthesis is one of the key points in MS alterations, a possible therapeutic approach would be to increase the sensitivity of adipocytes to insulin to increase their TG storage capacity [7].

*Tecoma stans* (L.) Juss. Ex Kunth. has been used as a medicinal plant in a traditional way since ancient times until nowadays in Mexico, Guatemala, and El Salvador, mainly as an antidiabetic treatment [8,9]. This species has also been studied using several different in vitro and in vivo models related to DM and obesity, has shown hypoglycaemic and antihyperglycaemic effects, TG and cholesterol reduction, as well as in vitro pancreatic lipase inhibition [10,11,12]. 

Considering that DM and MS share IR as one of their main pathophysiological disorders, and hyperglycaemia is present in both cases most of the time, the use of *T. stans* might have a positive effect on metabolic syndrome; however, its effects have not been evaluated in a model that tries to simulate the complexity of metabolic syndrome. Thus, the aim of this study was to evaluate the effects of the hydroalcoholic extract and the organic fraction of *T. stans* on some of the characteristic alterations of MS induced in a murine model by a hypercaloric diet, and to chemically characterize these treatments. The hydroalcoholic extract was obtained from leaves of *Tecoma stans* Juss. Ex Kunth, commonly known as yellow elder, using hydroalcoholic maceration (TsHAE). This extract was then subjected to liquid–liquid separation which produced an organic fraction (TsOF) which was then further fractionated, using silica gel chromatography, into four subfractions: TsSf1, TsSf2, TsSf3, and TsSf4, which contain different amounts of luteolin, apigenin, chrysoeriol, and caffeoyl derivative compounds. TsHAE, TsOF, and subfractions were proved in a hypercaloric diet murine model and compared to the two drugs Metformin and Orlistat for their effects on basal glycaemia, glucose tolerance, insulin resistance, triglycerides, body weight (BW), mesenteric adipose tissue weight, and blood pressure. These parameters are associated with the most common MS alterations: hyperglycaemia, hypertriglyceridaemia, obesity, and hypertension.

## 2. Results

### 2.1. Chemical Characterization

HPLC analysis of the TsHAE and the TsOF (Figure 1) showed that there were at least four main compounds in the TsHAE, with retention times (RTs) of 13, 17, 18, and 25 min. The first three of those compounds were also the most abundant in the TsOF. In addition, in the TsOF, there were four compounds with approximate RTs between 9 and 10 min, and there was no 25-min peak in this fraction. 

Comparison of the RTs and UV spectra of the compounds with those previously described for this species and other commercial standards indicated that the compounds with RT between 8 and 10 min present in the TsOF were mostly phenolic acids, specifically derivative esters of caffeic acid, while the peak with an RT of 13 min was luteolin and the peaks at 17 and 18 min were traces of apigenin and chrysoeriol, respectively.

The concentration of the identified flavonoids in the TsHAE and the TsOF was calculated using chromatographic profiles of commercial standards and showed that the TsHAE contained 0.6% luteolin, 0.5% apigenin, and 0.4% chrysoeriol, while the TsOF contained 3.7% luteolin, 0.9% apigenin, and 0.8% chrysoeriol.

HPLC analysis also allowed us to identify that TsSf1 contained luteolin as well as a caffeoyl derivative compound observed at 13.9 min, while TsSf2 contained a significant amount of luteolin, with slightly lower contents of apigenin and chrysoeriol (Figure 1). TsSf3 contained a high amount of luteolin and a caffeoyl derivative compound observed at 8.9 min, while the main component in TsSf4 was the same caffeoyl derivative observed at 8.9 min, and traces of other compounds with RT between 8 and 10 min were also present. 

A summary of the flavonoid content of TsHAE, TsOF, and subfractions 1 to 4 is shown in Table 1. The structures of these flavonoids are shown in Figure 2.

### 2.2. Hypercaloric Diet Model

Until week 10, no treatment was administered to any group. The administered treatments from week 10 were vehicle (Basal and MS groups), orlistat 40 mg/kg (Orlistat), metformin 100 mg/kg (Metformin), or 25 mg/kg of the different treatments obtained from *T. stans*.

#### 2.2.1. Insulin Resistance Parameters

The basal blood glucose concentrations in Table 2 show no effects of the controls or treatments during the first five weeks of administration. However, after 10 weeks of treatment, the TsOF- and TsSf1-treated groups had significantly lower blood glucose concentrations than the MS group and were even lower than that of the group receiving orlistat. The administration of metformin at 100 mg/kg did not show significant effects compared to the MS group.

TyG index values in the TsHAE, TsOF, TsSf1, TsSf4, and especially in the TsSf2 group were significantly lower than the values in the MS group, comparable to the orlistat group, and greater than that of the metformin group.

In the glucose tolerance test (GTT), the area under the curve (AUC) was higher in the MS group compared to the Basal group, but not in the rest of the groups, as shown in Table 2. Metformin, TsHAE, TsOF, and TsSf2 showed the most marked reduction in the AUC, particularly due to a decrease in the 20-min hyperglycaemia peak of the curve, and were comparable to the behaviour of the Basal group (Appendix A).

#### 2.2.2. Body Weight Parameters 

The change in BW over 10 weeks of treatment is presented in Figure 3, and the final weights are shown in Table 2. 

Among the treatments, the TsSf1 and TsSf2 groups had lower BW than the group treated with orlistat. These fractions have a significant content of chlorogenic acid, luteolin, apigenin, and chrysoeriol, and a low content of high polarity compounds. TsHAE had a similar effect to orlistat on BW, while there was no decrease in BW with the administration of TsOF or metformin. These effects were not due to changes in the lengths of the mice, which did not differ significantly among the groups (data not shown).

The administration of the hypercaloric diet in the MS group caused a statistically significant increase in the relative weight of mesenteric adipose tissue (MAT) compared to the control group, as shown in Table 2. TsOF did not reduce the amount of MAT. All the experimental treatments (particularly the subfractions) reduced the adiposity index, which represents total adipose tissue.

#### 2.2.3. Atherogenic Dyslipidaemia

As seen in Table 2, the TG levels were statistically different in all the groups that received the subfractions (87.8–93.6 mg/dL) with respect to the MS group (177.7 mg/dL), while this statistically significant difference was not observed with the administration of the rest of the treatments.

Although the composition of the diet was not expected to have a marked effect on blood cholesterol concentrations, there was a subtle increase in the MS group that was effectively controlled by most of the treatments, except TsSf4, the only fraction that did not have an important luteolin content.

#### 2.2.4. Hypertension

Both SBP and DBP were significantly higher in the hypercaloric diet group than in the Basal group, with mean differences of 20 and 12 mmHg, respectively, as shown in Table 2. This increase in blood pressure was not observed with the experimental treatments, with Orlistat, Metformin, TsHAE, TsOF, TsSf1, and TsSf4 being statistically equal to the Basal group. 

Although, in groups treated with TsSf2 and TsSf3, SBP was statistically equal to the Basal group. TsSf2 administration did not reduce DBP, while TsSf3 administration did decrease DBP but not as much as the other treatments. Considering that luteolin was the major compound in the less active fractions TsSf2 and TsSf3, it therefore seems that this flavone does not participate in the antihypertensive activity of *T. stans*. On the other hand, the caffeic acid derivative (RT of 13.9 min) and the other minor related compounds (RTs between 9 and 10 min) were identified in the active treatments. 

## 3. Discussion

The hypercaloric diet model presented in this work resembles, at least in part, the complexity of MS by presenting its main alterations simultaneously. 

The fact that only TsOF and TsSf1 achieved a decrease in glycaemia compared to the MS group suggests that the less polar compounds of *T. stans*, consisting of phenolic acid-type compounds, could improve this parameter. Some phenolic acids have well-described hypoglycaemic activities, for example, the effect of caffeic acid in mice with alloxan-induced DM [13], or the effect of ferulic acid in mice with a hypercaloric diet [14]. Chlorogenic acid, a compound previously identified in *T. stans*, also has multiple reports of glucose-lowering and/or glucose metabolism-enhancing activity [15,16,17]. 

The AUC of the GTT showed an improvement for all treatments, with TsHAE and TsSf2 being the most active. These treatments contain principally polyphenolic compounds such as luteolin, apigenin, and chrysoeriol. This suggests that flavone content is related to postprandial hyperglycaemia reduction. 

Glucose tolerance improvements could be due to α-glucosidase inhibition, previously reported for this species and proposed as the main antidiabetic mechanism [11]. On the other hand, Alonso-Castro et al. described the ability of this species to increase glucose uptake in adipocytes suggesting an insulin signalling pathway activation and TNF-α-induced insulin resistance reversion as the main mechanisms of action of this antidiabetic effect. This activity could also contribute to the effects observed in this work [18]. However, other antihyperglycemic mechanisms of action have not been investigated.

IR improvement was observed with most treatments, especially TsSf2, and is mainly due to the reduction in TG in serum. This effect has been previously reported with the administration of this species [11], as well as with the administration of apigenin (a compound previously identified in this species) in a murine model of obesity [19] and a model of DM [20].

Flavonoid-rich treatments (TsHAE, TsSf1, and TsSf2) had better effects on obesity than high-polarity compound-rich treatments (TsOF, TsSf3, and TsSF4). The flavonoids luteolin and apigenin have been reported to decrease BW in different pharmacological models [21,22,23], as well as chlorogenic acid [24], a compound previously reported for this species. While chrysoeriol has not demonstrated this effect in vivo, it has been shown to inhibit pancreatic lipase inhibition in vitro [25].

Although MAT is much less reported in studies of obesity-related diseases than total and epididymal adipose tissue, it is particularly important in the pathophysiology of MS because it is part of the visceral adipose tissue, which is directly involved in the exposure of hepatocytes to toxic metabolites contained in adipocytes, such as TG, AGL, and inflammatory cytokines, which promote the state of IR and the inflammation characteristic of MS [26]. Thus, finding a decrease in the weight of this tissue with the administration of most *T. stans* treatments is encouraging in the attempt to improve the general state of MS.

The decrease in body weight observed with the administration of the subfractions of *T. stans* has a behaviour like that observed with Orlistat, and the final weight, AI, and TAM were not lower than those observed in the Basal group. No changes were observed in the behaviour and food consumption of the mice (data not shown), suggesting that this effect is due to some non-aggressive antiobesity mechanism. However, the toxicity of these treatments, as well as that of pure luteolin, cannot be ruled out, and it is necessary to conduct a toxicological evaluation and a dose–response study.

Even though TsSf2 was indisputably the best treatment in terms of IR, atherogenic dyslipidaemia, and obesity parameters, it was the least active in reducing blood pressure, since it did not show diastolic blood pressure improvement. High blood pressure is one of the most important alterations to treat in MS, since it is one of the most frequent components in MS, related with great cardiovascular morbidity [27] and mortality [28].

Taking all of our findings into consideration, *T. stans*, and especially the TsOF, could be a suitable alternative for the treatment of MS. It improved most of the MS-related parameters evaluated in the present study, including IR, obesity, atherogenic dyslipidaemia, and arterial hypertension. In addition, it could improve the pro-inflammatory status characteristic of MS by achieving a significant reduction in MAT.

The liquid–liquid separation of TsHAE increased the flavonoid content in TsOF; luteolin changed from 0.6% to 3.7%, apigenin from 0.5% to 0.9%, and chrysoeriol from 0.4% to 0.8%. Although an increase in the concentration of luteolin in *T. stans* treatments seems to highly improve its effectiveness in terms of IR, it has no effect on hypertension. Considering that hypertension is a condition that directly impacts mortality rates associated with MS, luteolin-standardized *T. stans*-based phytomedicine appears to be preferable to the administration of isolated luteolin for MS treatment.

## 4. Materials and Methods

### 4.1. Plant Material 

*Tecoma stans* (L.) Juss. ex Kunth, commonly known as “yellow elder” in English and “tronadora” or “nixtamaxochitl” in Mexico, were collected in Cuernavaca, Morelos, Mexico (18°58′31.5″ N, 99°10′56.2″ W) and authenticated by Prof. Abigail Aguilar-Contreras. A voucher of plant material was deposited under accession number IMSSM-14,499 at the Medicinal Plant Herbarium of the Instituto Mexicano del Seguro Social (IMSS, Mexico City, Mexico). 

### 4.2. Phytochemical Procedures

Fresh leaves were washed and dried under dark conditions at room temperature and then milled to 4–6 mm. Ground material (1.15 kg) was extracted with a 60% ethanol solution to obtain 289 g of hydroalcoholic extract TsHAE. This complete extract was then subjected to liquid–liquid separation with a water/ethyl aetate immiscible mixture. The low-pressure distillation of the ethyl acetate allowed us to obtain 22 g of the organic fraction TsOF. 

The TsOF was subjected to chemical fractionation using open column chromatography, with silica gel 60 (109385, Merck KGaA, St. Louis, MO, USA) as the stationary phase and a hexane/ethyl acetate gradient system as the mobile phase. To select the most representative sub-fractions of TsOF—TsSf1, TsSf2, TsSf3, and TsSf4—thin layer chromatography was used, using aluminium sheets with silica gel 60 RP-18 F254s (105560, Merck KGaA). HPLC analysis was also used to monitor the chemical fractionations. HPLC was performed on Waters brand apparatus fitted with a Waters 2996 UV (900) photodiode array detector at 280 nm, using Empower 3 software and a SUPELCOSIL packed column (Supelco, St. Louis, MO, USA. LC-F^®^, 25 cm × 4.6 mm; 5 µm), with a trifluoroacetic acid/acetonitrile system as the mobile phase.

The 4 sub-fractions were evaluated in the murine model, and the HPLC chromatogram of the most active sub-fraction, TsSf2, was compared with the HPLC chromatograms of commercial standards (Sigma-Aldrich, St. Louis, MO, USA) of two previously identified compounds—apigenin and luteolin—and with a previously obtained and identified mixture of apigenin and chrysoeriol [25]. 

Based on the analysis of the chromatographic profiles of the commercial standards at 12.5, 25, 50, 100, and 200 μg/mL concentrations, calibration curves were built with the area under the curve of the peaks corresponding to the concentration of each compound. We used these calibrations to obtain y = mx + b form equations. The concentration of each compound in TsHAE and TsOF was then calculated using the acquired equation after obtaining the AUC of the peaks corresponding to these compounds.

### 4.3. Experimental Animals and MS Induction Model

The present study was approved by the Research Ethics Committee of the Mexican Social Security Institute with registration number R-2019-785-088 and was performed according to international, national, and institutional rules considering animal experiments, clinical studies, and biodiversity rights. The handling and care of the animals was conducted in accordance with internationally accepted procedures (Official Mexican Standard NOM-062-ZOO-1999, technical specifications for the production, care, and use of laboratory animals). 

Male C57BL6J mice were kept at 25 ± 3 °C under a 12 h:12 h light: dark cycle. Food and water were available ad libitum. When newly weaned, the mice were randomly divided into study groups of ten individuals (*n* = 10). Upon reaching a BW of 25 to 30 g (between the fifth and seventh week), they received either a standard diet (2028S, 18% Protein Rodent Diet Harland Tekland) or ad libitum hypercaloric diet (following the method of [29]) for 10 weeks. Animals receiving the hypercaloric diet that achieved a ≥ 15% BW increase compared to the control group continued to receive the hypercaloric diet for 10 more weeks, together with the oral treatment corresponding to their group, according to Table 3.

### 4.4. Evaluated Parameters

BW was measured weekly using a digital balance. After euthanasia, MAT, epididymal adipose tissue, subcutaneous adipose tissue, and perirenal tissue were obtained and weighed. The MAT relative weight was calculated by dividing it by the BW.

Each of these weights were summed to calculate the total adipose tissue (TAT) content, which was then used to calculate the adiposity index (AI) through the formula:AI (%) = (TAT weight/Body weight) × 100.(1)

Cholesterol was evaluated after 4 h of fasting in a blood sample using the Roche Accutrend Plus measuring device.

The concentration of glucose and TG was determined from serum samples obtained using the centrifugation of blood drawn from the infraorbital sinus under deep anaesthesia at the end of the experiments (week 20). Concentrations were determined using the BioSystems clinical chemistry kits: GLUCOSE OXIDASE/PEROXIDASE (11803) and GLYCEROL PHOSPHATE OXIDASE/PEROXIDASE (11828) following the manufacturer’s instructions.

The TyG index was used to indirectly detect IR [30] based on the concentration of glucose and TG using the formula:TyG = Ln(Triglycerides*Glucose)/2.(2)

A glucose tolerance test was conducted at the end of the experiment by measuring blood glucose levels after 4 h of fasting and at 15, 30, 60, and 120 min after the oral administration of the corresponding treatment and a glucose solution at 2 g/kg. With the data obtained, a graph was constructed, and the AUC was calculated.

Blood pressure was measured at the end of the experiment using a non-invasive method with a LETICA brand LE 5002 STORAGE PRESSURE METER. The animals were sedated per treatment group with a low dose of pentobarbital (10 mg/kg) intraperitoneally and warmed to 35 ± 2 °C for 5 min, and then, an insufflator ring attached to a transducer was placed at the base of the tail (Biopac System MP150, Goleta, CA, USA).

For each mouse, ten consecutive stable measurements were recorded one minute apart and averaged to obtain the systolic and diastolic pressure.

### 4.5. Data Processing and Statistical Analyses

For each data series, the mean and standard error of the mean (SEM) were calculated. The results in the graphs and tables are expressed as the mean ± SD.

Differences among groups were detected using an analysis of variance test with repeated measures (ANOVA-RM) and a Tukey post hoc test.

Differences were considered significant at *p* < 0.05. GraphPad Prism 9 software was used for graphing and statistical analysis.

## 5. Conclusions

A hypercaloric diet caused a metabolic syndrome-like state, which improved with the administration of hydroalcoholic extract and an organic fraction and subfractions of *T. stans*. The main compounds present in the active subfractions of *T. stans* were caffeate-like compounds, luteolin, apigenin, and chrysoeriol. An improvement in the alterations induced by a hypercaloric diet was observed by the administration of the *T. stans* subfractions; however, a better response was observed with the organic fraction, which contains 3.7% luteolin, 0.9% apigenin, and 0.8% chrysoeriol. As perspectives of this work, it is proposed to elucidate the mechanisms of action of the main compounds of *T. stans* in the different processes related with the physiopathology of each metabolic alteration of the metabolic syndrome.

## Figures and Tables

**Figure 1 plants-11-01794-f001:**
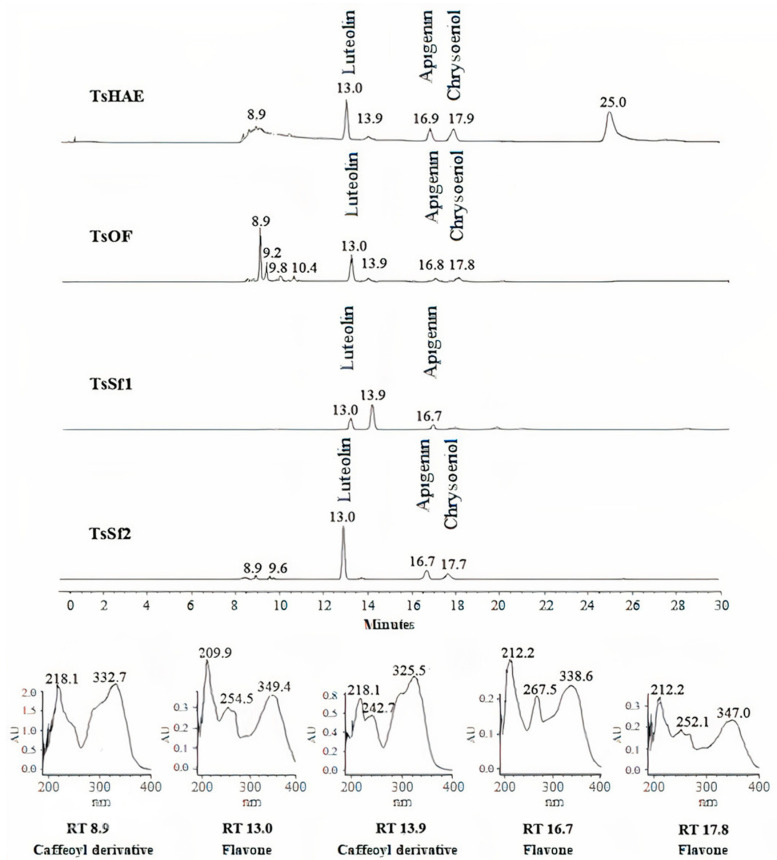
HPLC chromatogram comparison of hydroalcoholic extract TsHAE, ethyl acetate fraction TsOF, and subfractions TsSf1 and TsSf2 and UV spectra of major compounds.

**Figure 2 plants-11-01794-f002:**
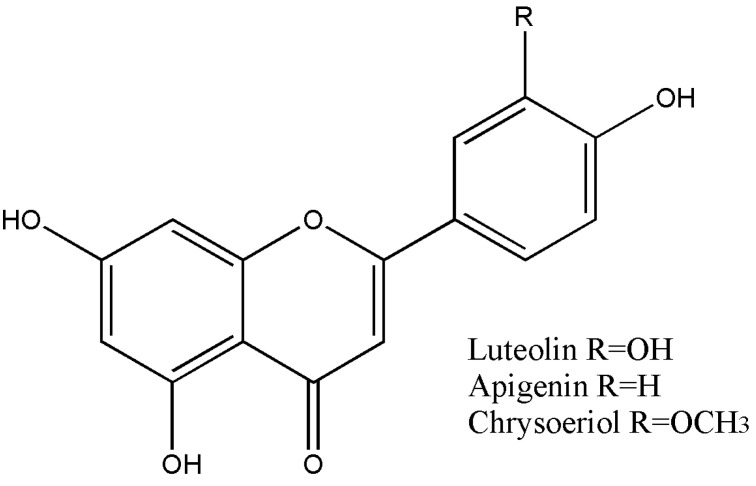
Structures of the compounds identified in *T. stans*: luteolin, apigenin, and chrysoeriol.

**Figure 3 plants-11-01794-f003:**
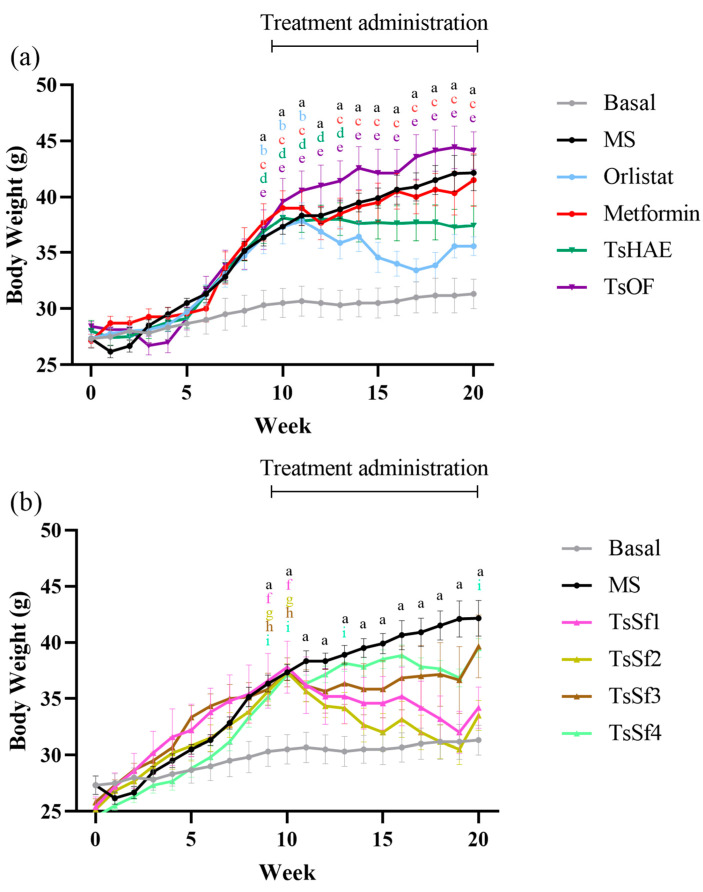
Change in body weight over time in mice before and during treatment. (**a**) Effect of hydroalcoholic extract and organic fraction of *T. stans* compared to control groups. (**b**) Effect of subfractions obtained from the organic fraction of *T. stans* (TsSf1, TsSf2, TsSf3 and TsSf4) compared to control groups. Data are expressed as mean ± SEM, *n* = 6 in each group. One-way ANOVA-RM, followed by Tukey’s test. ^a^ *p* < 0.05 by comparison of MS with Basal. ^b^ *p* < 0.05 by comparison of Orlistat with Basal. ^c^ *p* < 0.05 by comparison of Metformin with Basal. ^d^ *p* < 0.05 by comparison of TsHAE with Basal. ^e^ *p* < 0.05 by comparison of TsOF with Basal. ^f^ *p* < 0.05 by comparison of TsSf1 with Basal. ^g^ *p* < 0.05 by comparison of TsSf2 with Basal. ^h^ *p* < 0.05 by comparison of TsSf3 with Basal. ^i^ *p* < 0.05 by comparison of TsSf4 with Basal.

**Table 1 plants-11-01794-t001:** Content of identified flavonoids found using HPLC analysis in *Tecoma stans* treatments.

Treatment	Compound	%
TsHAE	Luteolin	0.65 ± 0.02
	Apigenin	0.49 ± 0.01
	Chrysoeriol	0.39 ± 0.01
TsOF	Luteolin	3.70 ± 0.02
	Apigenin	0.91 ± 0.05
	Chrysoeriol	0.77 ± 0.05
TsSf1	Luteolin	4.07 ± 0.06
	Apigenin	1.52 ± 0.14
	Chrysoeriol	1.18 ± 0.05
TsSf2	Luteolin	86.12 ± 0.41
	Apigenin	12.25 ± 0.24
	Chrysoeriol	1.08 ± 0.01
TsSf3	Luteolin	30.86 ± 0.23
	Apigenin	3.74 ± 0.06
	Chrysoeriol	1.95 ± 0.05
TsSf4	Luteolin	1.03 ± 0.03
	Apigenin	0.16 ± 0.01
	Chrysoeriol	0.25 ± 0.01

**Table 2 plants-11-01794-t002:** Effect of different parameters on mice fed with standard diet (Basal group) or hypercaloric diet (MS, Orlistat, Metformin, TsHAE, TsOF, TsSf1, TsSf2, TsSf3, and TsSf4 groups) after ten weeks of treatment with vehicle (in groups Basal and MS), orlistat 40 mg/kg (Orlistat), metformin 100 mg/kg (Metformin), and 25 mg/kg of different treatments obtained from *Tecoma stans*: hydroalcoholic extract (TsHAE), organic fraction (TsOF), subfraction 1 (TsSf1), subfraction 2 (TsSf2), subfraction 3 (TsSf3), and subfraction 4 (TsSf4).

Parameter/ Group	Glucose (mg/dL)	TyG Index	GTT AUC	BW (g)	AI (%)	MAT (mg/g)	TG (mg/dL)	Cholesterol (mg/dL)	SBP (mmHg)	DBP (mmHg)
Basal	65.3 ± 1.8 #	4.47 ± 0.05 #	19,727 ± 848 #	31.3 ± 1.3 #	5.4 ± 0.77 #	2.7 ± 0.9 #	91.4 ± 15.1 #	158.1 ± 1.6 #	89.9 ± 2.4 #	61.6 ± 1.7 #
MS	221.5 ± 14.6 *	5.16 ± 0.05 *	28,140 ± 966 *	42.2 ± 1.6 *	20.7 ± 0.95 *	22.8 ± 3.7 *	177.7 ± 14.7 *	165.3 ± 1.2 *	110.3 ± 3.5 *	73.7 ± 1.8 *
Orlistat	177.2 ± 12.2 *	**4.81 ± 0.05**	22,903 ± 545 #	**35.6 ± 0.8**	11.1 ± 0.88 *#	10.2 ± 1.3 #	101.9 ± 5.8	**160.1 ± 1.8**	87.4 ± 1.5 #	**53.0 ± 1.0** *#
Metformin	178.0 ± 9.8 *	5.01 ± 0.05 *	**18,996 ± 482** #	41.5 ± 2.3 *	16.5 ± 0.87 *	20.0 ± 3.2 *	130.2 ± 8.7	**153.4 ± 1.1** #	86.8 ± 1.3 #	60.2 ± 2.3 #
TsHAE	166.5 ± 12.5 *	4.93 ± 0.10 *	**20,739 ± 1202** #	37.4 ± 1.8	13.1 ± 1.96 *#	10.3 ± 2.2 #	125.0 ± 21.4	163.7 ± 0.9 *	91.8 ± 3.9 #	**56.4 ± 4.5** #
TsOF	**140.3 ± 13.4** *#	**4.78 ± 0.14** #	22,730 ± 1173 #	44.1 ± 1.7 *	14.6 ± 0.75 *#	15.7 ± 3.1 *	124.2 ± 19.5	**158.1 ± 0.3** #	86.6 ± 1.0 #	**52.8 ± 1.2** *#
TsSf1	**144.2 ± 5.7** *#	**4.72 ± 0.02** #	23,478 ± 580	**34.2 ± 1.8**	**7.1 ± 0.97** #	**4.4 ± 1.1** #	**87.8 ± 2.6** #	161.0 ± 1.3	**75.6 ± 0.7** #	58.8 ± 1.7 #
TsSf2	**165.6 ± 5.3** *	4.80 ± 0.03	**22,818 ± 392** #	**33.5 ± 1.3** #	**3.9 ± 1.14** #	**2.0 ± 0.9** #	**89.6 ± 4.8** #	160.8 ± 0.6	91.4 ± 3.4 #	74.9 ± 3.5 *
TsSf3	208.6 ± 14.5 *	4.93 ± 0.05 *	23,798 ± 389	39.7 ± 2.8	8.9 ± 1.43 #	9.4 ± 2.1 #	93.6 ± 4.3 #	161.0 ± 3.7	**81.1 ± 2.4** #	65.0 ± 2.3
TsSf4	200.0 ± 7.8 *	4.86 ± 0.03	24,609 ± 785	39.5 ± 0.8	**8.0 ± 0.82** #	**7.4 ± 1.0** #	**83.4 ± 2.2** #	166.0 ± 2.0 *	**75.1 ± 1.5** #	57.8 ± 2.1 #

GTT: glucose tolerance test; AUC: area under curve; BW: body weight; AI: adiposity index; MAT: mesenteric adipose tissue; TG: triglycerides; SBP: systolic blood pressure; DBP: diastolic blood pressure. Data are expressed as mean ± SEM, *n* = 6 in each group. One-way ANOVA-RM, followed by Tukey’s test. * *p* < 0.05 compared to the Basal group. # *p* < 0.05 compared to the MS group.

**Table 3 plants-11-01794-t003:** Groups in the murine metabolic syndrome induction model.

Group	Diet	Treatment in Weeks 10 to 20
Basal	Standard	Vehicle (5% tween 20)
MS	Hypercaloric	Vehicle (5% tween 20)
Orlistat	Hypercaloric	Orlistat 40 mg/kg
Metformin	Hypercaloric	Metformin 100 mg/kg
TsHAE	Hypercaloric	*T. stans* hydroalcoholic extract 25 mg/kg
TsOF	Hypercaloric	*T. stans* organic fraction 25 mg/kg
TsSf1	Hypercaloric	*T. stans* subfraction 1 25 mg/kg
TsSf2	Hypercaloric	*T. stans* subfraction 2 25 mg/kg
TsSf3	Hypercaloric	*T. stans* subfraction 3 25 mg/kg
TsSf4	Hypercaloric	*T. stans* subfraction 4 25 mg/kg

## Data Availability

Not applicable.

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
