# Peer review of "Effect of Tecoma stans (L.) Juss. ex Kunth in a Murine Model of Metabolic Syndrome"

_plants, 2022, doi:10.3390/plants11141794_

Round 1

Reviewer 1 Report

Reviewer comments and suggestions

The study evaluated the effects of Tecoma stans (L.) Juss. Ex Kunth. in a hypercaloric diet-induced metabolic syndrome model. The authors used the organic fraction obtained by a liquid-liquid separation from the hydroalcoholic extract of T. stans, and four subfractions of this organic fraction by open column chromatography were used to treat the mice for ten weeks. The study results showed favorable effects in most of the evaluated parameters with the administration of T. stans subfractions, though, the organic fraction and the hydroalcoholic extract observed better responses. The authors briefly suggested that T. stans treatments, a flavonoid-rich organic fraction, achieve an improvement of metabolic syndrome alteration.

The paper seems to be weak in terms of cited references and information provided in the material and methods section. Based on my view, below are the comments that need to be incorporated into the revised version of the manuscript. 

  1. In the abstract, Line 21-22 this was not important to write here in the abstract, it would be better if the authors describe the results obtained. 1-2 line conclusion is needed”
  2. Line 36, mellitus not need to be italic
  3. Line 84-85 The authors can also suggest the obese model or high-fat diet model why do they suggest MS model
  4. Line 136 lower or higher, the sentence needs to be completed
  5. For the hypertension section, The authors need to discuss all the parameters of MS if they considered these animals to have the syndrome. Better it would be high-fat diet/high-calorie diet used in the experiment 
  6. Line 149-150 is there was a possible reason for this
  7. Please modify the first para of the discussion. The three consecutive lines were not necessary
  8. Line 162-163 please explore the sentence
  9. Line 169-170 please explain the results in the discussion
  10. Line 172 Check the cited reference here in the text
  11. Line 176-178 Do the authors have histological analysis, please include
  12. Line 185-187 Do the authors have a reference for this. Line 186-187 needs to explore more
  13.  Line 200-205 These lines are not required
  14. Line 241 please mention the animal ethical approval number for conducting the study
  15. Please avoid the big sentence line 293-295. Line 298 The lines seem confusing. better to present in a professional way

Reviewer 2 Report

Interesting results, revisions have been suggested to make the paper easier to follow.

Reviewer 3 Report

The topic of the manuscript is interesting. However, the reviewer feels it only possess limited significance.  

1) Did the authors identify any new phyto-chemical?

2) Fig 2 b, TsSf1-4 caused significant body weight loss? Implying toxicity? 

3) What is the mechanism of action? This is most important. 

Reviewer 4 Report

please see the attached file

Round 2

Reviewer 1 Report

No more comments. All comments have been updated by the authors.

Reviewer 3 Report

The reviewer has no objection to accept to this manuscript. 

Reviewer 4 Report

The format of reference in the references sections should be followed the journal format and uniform, ex:

1. Gimenez-Ibanez, S.; Hann, D.R.; Ntoukakis, V.; Petutschnig, E.; Lipka, V.; Rathjen, J.P. AvrPtoB targets the LysM receptor kinase CERK1 to promote bacterial virulence on plants. Curr. Biol. 2009, 19, 423–429.

2. Bektas, Y.; Eulgem, T. Synthetic plant defense elicitors. Front. Plant Sci. 2015, 5, 804.

3. Quintana-Rodriguez, E.; Duran-Flores, D.; Heil, M.; Camacho-Coronel, X. Damage-associated molecular patterns (DAMPs) as future plant vaccines that protect crops from pests. Sci. Hortic. 2018, 237, 207–220.